# Preventing Punitive Violence: Implementing Positive Discipline in Everyday Parenting (PDEP) with Marginalized Populations in Bangladesh

**DOI:** 10.3390/ijerph20031873

**Published:** 2023-01-19

**Authors:** Christine A. Ateah, Laila Khondkar, Firozul Milon, Rasheda Rabbani

**Affiliations:** 1College of Nursing, Rady Faculty of Health Sciences, University of Manitoba, Winnipeg, MB R3T 2N2, Canada; 2Kennedy School of Government, Harvard University, Cambridge, MA 02138, USA; 3Bangladesh Environmental Lawyers Association, Dhaka 1205, Bangladesh; 4George & Fay Yee Centre for Healthcare Innovation, Rady Faculty of Health Sciences, University of Manitoba, Winnipeg, MB R3E 0T6, Canada

**Keywords:** physical punishment, emotional punishment, corporal punishment, parenting, violence, Positive Discipline in Everyday Parenting, prevention of child abuse, child maltreatment

## Abstract

Physical and other types of punishment remain common in Bangladesh, despite overwhelming evidence of their harm and worldwide efforts to decrease their use. One of the strategic priorities of Save the Children in Bangladesh’s Child Protection Program is to protect children from physical and humiliating punishment in homes, schools, and other settings. Save the Children in Bangladesh selected the Positive Discipline in Everyday Parenting (PDEP) Program to provide parents with alternatives to physical punishment that comply with human rights standards while strengthening relationships and understanding of child development. High-risk communities where children are particularly vulnerable were selected for this project. The PDEP program was delivered to 857 parents living in lower socioeconomic areas of Bangladesh, including ethnic minority groups, and parents living in urban slums of Dhaka and rural brothel areas. Due to the low levels of education of the participants (almost two-thirds of participants had not completed elementary school), simplified pre and posttests were utilized. Following program completion, parents’ approval of both physical punishment and punishment in general declined; they were less likely to view typical parent-child conflicts as intentional misbehavior and were less reactive to frustration. In addition, parents indicated an increased understanding of the positive discipline and more confidence in their parenting skills. Before taking PDEP, 64% of the parents often felt like they just did not know what to do as a parent, compared to 34% following program completion. PDEP demonstrated the potential to decrease the use of physical and humiliating punishments by parents living in high-risk communities in Bangladesh.

## 1. Introduction

Over the last 30 years, findings on the negative effects of physical or corporal punishment have grown exponentially [1,2,3], as has the recognition of the personhood of children and their right to live without violence. There is overwhelming evidence of the harms of corporal punishment. For example, a review of over 300 studies was found to demonstrate associations between corporal punishment and numerous negative outcomes. These negative outcomes include direct physical harm; poor moral internalization, increased antisocial behavior; increased aggression in children; adult perpetration of violence; mental and physical health issues in children; and damage to the parent-child relationship [4]. Following the United Nations (UN) 1989 [5] adoption of the Convention on the Rights of the Child (CRC) and the resultant consideration of how children have been treated, views of childhood are shifting on a global level. To date, 65 countries have abolished all physical punishment of children [6]. In addition to seeking law reform, governments and other organizations are searching for ways to support parents in promoting children’s healthy development. 

### 1.1. Bangladesh Context

Bangladesh is one of the most densely populated countries in the world and is classified as a medium development country by the United Nations Development Programme [7] using the Human Development Index (HDI). The HDI is a summary measure of average achievement in key dimensions of human development: life expectancy, education, and standard of living. Physical punishment of children remains legal in the home, alternative care settings, daycare, penal institutions, and as a sentence for a crime. A Bangladesh Supreme Court public interest litigation ruling in 2011 against corporal punishment in schools still requires confirmation in legislation [8]. According to the Multiple Indicator Cluster Survey, 89% of children between the ages of 1–14 years in Bangladesh experienced violent discipline (physical and/or psychological aggression) in the month prior to the survey [9]. One of the strategic priorities of Save the Children in Bangladesh’s (SCB) child protection program is to protect children from physical and humiliating punishment in homes, schools, workplaces, and communities. This priority has been addressed since 2014 through the implementation of Positive Discipline in Everyday Parenting (PDEP).

### 1.2. Positive Discipline in Everyday Parenting (PDEP)

PDEP was developed as a primary violence prevention program with the aim of reducing the physical punishment of children. It grew out of an academic-NGO partnership between Joan Durrant (University of Manitoba) and Save the Children, an international non-governmental organization (NGO) that promotes children’s rights. In 2007, with the support of Save the Children Sweden, Durrant wrote a book for parents that describes a model of parent-child conflict resolution based on trust, attachment, communication, and validating children’s perspectives that can be applied in a wide range of situations [10]. PDEP’s theoretical approach is based on the theory of planned behavior [11] and describes how behavioral beliefs (e.g., attitude toward physical punishment); normative beliefs (e.g., perceiving parent-child conflict as misbehavior; and control beliefs (e.g., self-efficacy) affect intention and behavior. Therefore, reducing attitudes toward approval of physical punishment, increasing knowledge of normative child behavior, and increasing parental self-efficacy are key to the PDEP program. A facilitator training model was developed, and a facilitator’s manual [12] was produced to support trainees in their delivery of the program to parents through community agencies, schools, health centers, and other community-based agencies. Facilitators typically are parent educators, NGO staff, teachers, childcare workers, and other professionals working directly with families. The program is organized into eight two-hour sessions, plus a follow-up session, for groups of 10 to 18 parents.

PDEP has been implemented in over 30 countries to date. The program was designed to help parents move from external control strategies (e.g., physical punishment, time-out, removal of privileges) to mentorship and conflict resolution. Designed to capture the fundamental principles of caregiving that promote developmental health throughout childhood and adolescence, the PDEP framework consists of five components: (1) focusing on long-term parenting goals; (2) creating a learning environment in which children feel physically and emotionally safe (“warmth”); (3) scaffolding children’s learning (“structure”); (4) understanding children’s perspectives across the developmental trajectory, and (5) approaching discipline as problem-solving rather than punishment. The program takes parents through these components in sequence, with each building on the previous ones.

A series of interactive exercises help parents to understand the rationale for the approach, gain insight into their children’s thinking, and generate constructive, nonviolent solutions on their own. Four of the eight sessions take parents through the typical developmental pathway, from birth to adolescence, with a focus on developmental changes across the age span, such as attachment and the drive for autonomy. Brain development is emphasized to help parents understand the impact of stress and aggression on children and how their own emotion regulation can facilitate the growth of self-regulation in their children [13].

PDEP also helps parents understand the fundamental principles of children’s rights, including the right to express their perspectives and to have them heard. PDEP aims to reorient parents from relationships with their children that are based on power and control to relationships based on cooperation, reciprocity, and mutual respect.

Early in the program, parents work in small groups on exercises aimed at normalizing parenting stress, parent-child conflict, and stress-based reactions. As the program proceeds, parents continue to engage in small group activities designed to re-frame children’s behavior from ‘bad’ to ‘developmentally normative’ and to help parents recognize their own strengths and capacities. Eventually, parents brainstorm problem-solving responses to conflict using the PDEP framework.

Pre- and post-program questionnaires are completed at both the parent program and facilitator training levels. Findings from a sample of 321 parents who participated in the program in 14 sites across Canada [11] indicate that PDEP has promise as a violence prevention program. For example, following completion of the PDEP program, 82% of parents believed that they could now solve most of their parenting challenges compared to 62% at the pretest (*z* = −2.92, *p* < 0.001). In the self-assessment of the impact of PDEP following the program, 84% believed that PDEP would help them use less physical punishment; and more than 90% believed that PDEP would help them control their anger, understand their children’s feelings and build stronger relationships with their children [11]. In another study of parents who completed the PDEP parent program (*N* = 172) in the Asia-Pacific region (Australia, Japan, and the Philippines), Durrant et al. demonstrated similar results [13]. Although there were some individual country differences, in all three countries, participants’ behavioral beliefs shifted following the program, and their approval of physical punishment declined, regardless of gender, education, or the number of children in the family. Parents also became less likely to attribute typical child behavior to intentional misbehavior. At least 75% of parents in each country reported a perceived increase in feeling better prepared to respond constructively to conflict with their children. Given the higher prevalence of the use of physical punishment in those countries compared to Canada [13,14], this study provided a stronger test of the program’s target impacts namely, reducing approval of physical punishment, reducing attributions of typical child behavior to intentional misbehavior and increasing parents’ self-efficacy in responding to conflict, without physical or emotional punishment.

### 1.3. Purpose

The purpose of the current study was to examine whether Bangladeshi parents from marginalized populations who have taken the PDEP program demonstrate evidence of change in their support for physical punishment and punishment in general, feelings of self-efficacy as parents, understanding of the positive discipline, understanding of typical parent-child conflicts, and reactivity to frustration. Given the efforts reported to make PDEP universal and adaptable [11,13] and the positive findings reported by parents in Canadian and international locations other than Bangladesh, we anticipated similar outcomes. Specifically, we hypothesized that parents who completed the PDEP program would report: (1) decreased support for physical punishment; (2) decreased support for punishment in general; (3) increased parenting self-efficacy; (4) increased understanding of positive discipline; (5) decreased attributions of typical parent-child conflicts to intentional child behavior; and (6) decreased reactive responses (frustration) to conflict with their children.

## 2. Materials and Methods

### 2.1. Participants

The sample was comprised of 857 parents who completed PDEP parent programs in urban slums in Dhaka, rural brothel areas of Faridpur and Rajbari, and among two ethnic minority groups (Bowm, Keyang) in Bandarban. Parents from ethnic minority groups and parents in urban areas and brothels are marginalized due to poverty, social exclusion, and discrimination. Save the Children in Bangladesh focuses on protecting the rights of the most marginalized groups, including families from these particular groups noted above. Participants for this study were a convenience sample of parents from marginalized groups who had accessed services from partners of Save the Children and enrolled in PDEP programs offered in their communities.

In brothel areas, participants included sex workers, pimps, and other residents. Historically sex workers have been marginalized as a group and face a lot of stigma and discrimination. Sex workers, people living in urban slums, and members from ethnic minority groups have lower socioeconomic status in general, few opportunities for decent work, lack quality housing and have limited access to water and sanitation facilities. Due to all these factors, stress on caregivers in these communities can make children more vulnerable to physical and humiliating punishment [15].

Save the Children in Bangladesh has been working in these communities for more than a decade. In partnership with local NGOs, Save the Children in Bangladesh implemented interventions to develop parental and community awareness on protecting children, facilitated Community Based Child Protection Committees so that communities can perform responsibilities to protect children, and developed referral linkages with government and NGO service providers (e.g., health, education, social service, legal). Save the Children in Bangladesh also implemented education initiatives for children so that they become aware of their rights and can protect themselves from violence. PDEP was implemented in order to strengthen the capacity of parents to raise children in a nonviolent way.

The sample demographics are presented in Table 1. Approximately one-third (35.94%) of the sample were aged 29 years or less (only 2.2 % of the sample were under 20 years of age). Almost two-thirds (63%) of the participants had not completed elementary school, and none had post-secondary education. Most participants were female (60.91%). About one-third of participants had one or two children, and another third had five or more.

### 2.2. Measures

Due to participants’ low level of education, simplified versions of the standard PDEP pre- and post-program questionnaires were administered (see [11] for a description of the standard questionnaires and their development). The simplified pre and post-program questionnaires consisted of a subset of 14 items drawn from the standard questionnaires. Each of the six research constructs was represented by two to three statements based on the standard questionnaire, resulting in a 14-item pre- and post-test questionnaire with 4-point levels of agreement. As it is recognized that it is important to assess the relevance of the program to populations, additional questions were added to the posttest questionnaire to determine participants’ satisfaction with the program, the parent book, the program facilitator, length of the program, and if they would recommend the program to other parents.

Participants rated their agreement with each statement on a four-point scale (1 = strongly disagree, 2 = mostly disagree, 3 = mostly agree, 4 = strongly agree). The Flesch Reading Ease score was 78.8, placing the simplified questionnaires in the ‘fairly easy to read’ category. The Flesch–Kincaid Grade Level was 4.4, indicating that the questionnaires can be easily understood by a US reader with 4.4 years of education [16].

The pre-program questionnaire was administered at the beginning of the first program session, prior to the delivery of any program content. Participants were first asked to indicate their sex, age, highest level of education, and number of children. They were then presented with 14 statements measuring the constructs of interest. Support for physical punishment was operationalized by the strength of agreement that (1) Spanking is fine as long as the parent is not angry, and (2) Parents should have the right to decide whether to spank their children. Support for punishment, in general, was operationalized by the strength of agreement that (1) Children who are punished learn to behave better than children who aren’t punished, and (2) If parents don’t use punishment, their children will be spoiled. Parenting self-efficacy was operationalized as the strength of agreement that (1) Most people are better parents than I am (reverse-scored), and (2) I have the skills to be a good parent; and (3) As a parent, I often just don’t know what to do (reverse-scored).

Understanding positive discipline was operationalized by the strength of agreement that (1) “Positive discipline” means using punishments, such as taking things away from a child (reverse scored), and (2) “Positive discipline” usually means letting children do whatever they want (reverse scored). Attributions of typical child behavior to intentional misbehavior were operationalized by the strength of agreement that (1) Four-year-olds who interrupt adults are rude; (2) Babies cry in the middle of the night to make their parents angry; and (3) If a teenager says her parents’ rules are unfair, she should be told, “If you don’t like the rules you can move out.” Reactivity to frustration was operationalized by the strength of agreement with the statements: (1) When I argue with my child, I often say things I don’t mean; and (2) When my child does something I don’t like, I sometimes yell.

The post-program questionnaire was administered at the end of the final session after all program content was completed. It consisted of the same 14 statements as the pre-program questionnaire, which were rated by participants on the same four-point agreement scale. At the posttest, parents were also asked to indicate their level of satisfaction on a 4-point scale (very dissatisfied, satisfied, dissatisfied, strongly dissatisfied) regarding the overall program, the facility, the Positive Discipline parent book [10], the PDEP program exercises, the program leader, and the length of the program. In addition, participants were asked if they would recommend the program to other parents (yes, no, unsure).

### 2.3. Procedure

Approval for the study was obtained from the University of Manitoba’s Research Ethics Board prior to the commencement of data collection. All PDEP program facilitators were trained in the administration of the measures and in following ethical protocol. The low education levels of the participants necessitated some adaptations to standard program delivery. For example, in groups where many parents had very low literacy, the content was presented via pictures and roleplay rather than materials that required reading. These alternative delivery methods are clearly described in the PDEP Facilitator Manual [12], enabling program facilitators to use them in a consistent way across programs.

### 2.4. Analysis

The data were analyzed using SAS (9.4) and SPSS 27 [17]). First, descriptive statistics were calculated to examine the distributions of all variables. Second, Cronbach’s alpha was used to measure scale reliability. Third, because Cronbach’s alpha can be low when there are a small number of items, following Bartlett’s test of Sphericity for the strength of correlations, confirmatory factor analysis was conducted to examine dimensionality using principal component analysis to determine correlations between the questionnaire items and constructs.

Fourth, a univariable and multivariable linear mixed model [18] was used to test for a statistically significant change in average mean scores between pre and posttest after adjusting for demographic covariates (age, sex, education, and number of children). Missing cases were removed from the pretest, posttest, and all demographic variables. Model fit and assumptions were assessed via a visual assessment of residuals. Statistical significance was determined based on 95% confidence intervals and an alpha level of less than 0.05. Multiple comparison tests were applied where appropriate.

## 3. Results

Following the confirmatory factor analysis, because of a negative factor loading (−699), the item “I have the skills to be a good parent” was removed from the parenting self-efficacy construct. Although this item did not correlate with the self-efficacy construct, 86% of parents did agree or strongly agree with that statement following completion of the program, compared with 79% pre-program. The analysis shows there is only one component/factor for each construct with corresponding items. Therefore, it can be concluded that items are correlated with the correspondence construct. See Table 2 to view the items retained following principal component analysis and their factor loadings.

Our hypotheses for expected differences following the completion of the PDEP program were supported. Expected changes for all constructs were evident both before and following control for demographic variables (*p* < 0.0001). Following program completion, support for physical punishment (M = 3.66, ± SE 0.07) decreased (M = 2.33, ± SE 0.07), and for punishment in general (M = 3.99, ± SE 0.07) also decreased (M = 2.16, ± SE 0.07). Parenting confidence (efficacy) (M = 4.79, ± SE 0.08) increased (M = 6.18, ± SE 0.08), and understanding of positive discipline as neither punitive nor permissive (M = 3.77, ± SE 0.07) decreased (expected change) (M = 2.68, ± SE 0.07). Viewing of typical parent-child conflicts as problems (M = 4.50, ± SE 0.08) decreased (M = 3.03, ± SE 0.08), and experiencing reactive responses (frustration) to conflict with their children (M = 4.26, ± SE 0.08) decreased (M = 3.40, ± SE 0.08). See Table 3.

Parents rated the PDEP program very highly on the posttest; responses were “mostly satisfied” or “very satisfied” by almost all (98−99%) participants in all areas: overall program, facility, PDEP book, the program exercises, facilitators, and the length of the program. In addition, nearly all participants would recommend the program to other parents.

## 4. Discussion

The purpose of the current study was to examine whether Bangladeshi parents from marginalized populations who have taken the PDEP program demonstrate evidence of change in their support for physical punishment and punishment in general and change in the constructs of parenting self-efficacy, understanding of positive discipline, understanding of typical parent-child conflicts, and reactivity to frustration. All five hypotheses were supported. There were statistically significant changes in mean scores between pre and post-test after adjusting for demographic covariates (age, sex, education, and number of children).

The finding of decreased support for physical punishment and for punishment, in general, is an important finding since a positive attitude toward physical punishment has been found to be a primary predictor of its use [19]. Durrant et al. [11] also reported a decrease in attitudes toward physical punishment in their study of a Canadian sample. The PDEP program focuses on the understanding that punishment does not provide children with what they need to learn expected behavior, which is warmth (love) and structure (guidance and support), and actually is counterproductive.

Parents reported an increase in parenting self-efficacy. Parents were less likely than before taking the program to feel that other parents were better than themselves, or they often did not know what to do. As the program focuses on nonpunitive responses with their children, this finding supports that parents feel more confident in this new approach and in fact, did also demonstrate an increased understanding of the positive discipline. Parents reported decreased attributions of typical parent−child conflicts to intentional child behavior. This finding demonstrates that parents had a greater understanding that parent-child conflict is expected and natural and that applying punishments rather than problem-solving is counterproductive. Parents also reported a decrease in reactive responses (frustration) to conflict with their children. This finding is important because anger has been found to be a predictor of physical punishment [19]. Parents are less likely to feel anger if they are not attributing conflict as misbehavior and, therefore, less likely to respond with anger, yelling, and the use of physical punishment.

This study demonstrates that PDEP is applicable and useful to parents with low education levels and living in challenging situations. Previous studies reporting positive outcomes of PDEP program delivery [11,13] had incorporated the standard program and pre-and post-test questionnaires. Adaptations to the PDEP program, such as using a translator for Bengali-speaking parents; and pictorial and role play for key concepts as needed, likely contributed to successful program delivery in Bangladesh as described. PDEP has the potential to shift attitudes toward punishment and perceived parenting competency in challenging contexts. Parents who completed the program show positive indications that they no longer support physical or other punishments or feel it is the best option for responding to their children. Parents who are less likely to experience frustration during typical child behavior can reasonably be thought to respond to their children with more positive responses, as taught in the program. It is recommended that PDEP programming in Bangladesh continue and expand. In particular, it would be preferable if trainers/facilitators could be developed from local communities. Although the program evaluations were positive, it is always preferable to have local trainers who understand the community dynamics, programs, and supports available.

## 5. Limitations

An important limitation of this study is that parents completed the posttest at the end of the program but were not followed up to determine the longer−term effect. It is unknown how long their change of perspective will continue, and/or whether their change of attitudes and beliefs in fact will result in a decreased use of physical and other punishments. Ateah and Durrant [19] found that attitude toward physical punishment was the strongest predictor of its use with young children, so if parents in this study continue to no longer support the use of physical punishment, it is hopeful that they will use the problem solving approach of PDEP instead. However, future research should include further study of parent attitudes and behaviors with a post−posttest of parents’ behavior change. As PDEP is based on the theory of planned behavior [11] it will be important to evaluate how the shift in behavioral beliefs, normative beliefs and control beliefs affect parent behavior on a long-term basis.

## 6. Conclusions

Despite the harms of physical punishment, it continues to be commonly used toward children in many parts of the world, including Bangladesh. Changing long-standing practices of punishing children requires a new way of viewing children and child-rearing practices that promote healthy development. The PDEP program has shown promise as a violence prevention program. Save the Children in Bangladesh chose the PDEP program to be delivered to parents living in challenging conditions and with low education levels. Following completion of the PDEP program, most parents indicated a decrease in approval of both physical and other punishments. Parents reported they were less likely to view typical parent-child conflicts as problematic or to experience frustration with their children. In addition, parents indicated an increased understanding of positive discipline and more confidence in their parenting skills. Virtually all participants would recommend the program to other parents. PDEP has demonstrated the potential to decrease the use of punitive violence with children and provide support to parents, including those who experience challenges and marginalization.

## Figures and Tables

**Table 1 ijerph-20-01873-t001:** Demographic Characteristics N = 857.

Variables	*n* (%)
**Age (in years)**	
<29	308 (35.94)
30–39	296 (34.54)
>40	245 (28.59)
Missing	8 (0.93)
**Gender**	
Female	522 (60.91)
Male	329 (38.39)
Missing	6 (0.70)
**Number of Children**	
1–2	302 (35.24)
3–4	246 (28.70)
5+	279 (32.56)
Missing	30 (3.50)
**Highest Level of Education**	
Did not complete elementary school	542 (63.24)
Completed elementary school	141 (16.45)
Completed high school	109 (12.72)
Missing	65 (7.58)

**Table 2 ijerph-20-01873-t002:** Items retained following principal component analysis and factor loadings.

Factor	Factor Loadings	Cronbach’s Alpha	Bartlett’s Test of Sphericity
Support of physical punishment		0.48	<0.0001
Q10	0.814		
Q12	0.814		
Support for punishment in general		0.61	<0.0001
Q9	0.848		
Q14	0.848		
Parenting self-efficacy		0.24	<0.0001
Q6	0.753		
Q13	0.753		
Understanding of positive discipline		0.51	<0.0001
Q1	0.820		
Q5	0.820		
Understanding of typical parent-child conflicts		0.72	<0.0001
Q2	0.828		
Q3	0.813		
Q11	0.758		
Response to frustration		0.37	<0.0001
Q4	0.784		
Q7	0.784		

**Table 3 ijerph-20-01873-t003:** Pre- and posttest means and differences.

	Unadjusted for Demographic Variables Effect			Adjusted for Demographic Variables Effect		
Objective	Mean (±SE)	Diff (95%CI)	*p*	*N* *, Mean (±SE)	Diff (95%CI)	*p*
Support for physical punishment (*N* * = 840)				*N* * = 743	−1.33 (−1.49 to −1.18)	<0.0001
Baseline (*n* = 842) Post (*n* = 854)	3.83 (0.07) 2.46 (0.03)	−1.35 (−1.49 to −1.21)	<0.0001	3.66 (0.07) 2.33 (0.07)		
Support for punishment in general (*N* * = 839)				*N* * = 741	−1.83 (−1.99 to −1.66)	<0.0001
Baseline (*n* = 841) Post (*n* = 854)	4.18 (0.07) 2.30 (0.03)	−1.88 (−2.03 to −1.73)	<0.0001	3.99 (0.07) 2.16 (0.07)		
Parenting self−efficacy (*N* * = 855)				*N* * = 754	1.39 (1.20 to 1.58)	<0.0001
Baseline (*n* = 855) Post (*n* = 857)	4.80 (0.07) 6.19 (0.07)	1.39 (1.22 to 1.57)	<0.0001	4.79 (0.08) 6.18 (0.08)		
Understanding of positive discipline (*N* * = 852)				*N* * = 754	−1.08 (−1.24 to −0.92)	<0.0001
Baseline (*n* = 855) Post (*n* = 854)	3.80 (0.06) 2.74 (0.04)	−1.06 (−1.20 to −0.92)	<0.0001	3.77 (0.07) 2.68 (0.07)		
Understanding of typical parent−child conflicts (*N* * = 855)				*N* * = 754	−1.47 (−1.66 to −1.28)	<0.0001
Baseline (*n* = 855) Post (*n* = 857)	4.73 (0.09) 3.27 (0.03)	−1.46 (−1.63 to −1.29)	<0.0001	4.50 (0.08) 3.03 (0.08)		
Response to frustration (*N* * = 852)				*N* * = 753	−0.87 (−1.06 to −0.68)	<0.0001
Baseline (*n* = 853) Post (*n* = 856)	4.35 (0.07) 3.44 (0.06)	−0.91 (−1.07 to −0.74)	<0.0001	4.26 (0.08) 3.40 (0.08)		

* After deleting the missing numbers from the baseline, posttest, and all demographic variables, the baseline and posttest have the same numbers.

## Data Availability

Data are not publicly available due to ethical restrictions. The data presented in this study are available on request from the corresponding author.

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
