# Peer review of "Preventing Punitive Violence: Implementing Positive Discipline in Everyday Parenting (PDEP) with Marginalized Populations in Bangladesh"

_ijerph, 2023, doi:10.3390/ijerph20031873_

Round 1

Reviewer 1 Report

Preventing Punitive Violence: Implementing Positive Discipline in Everyday Parenting (PDEP) with Marginalized Populations in Bangladesh 

Streamline the abstract so that more of the current study becomes evident in the section. 

1.2 - Use the full name instead of PDEP 

This section requires reworking and rewriting – the authors should promote and rationalise the strategy proposed by Durrant (2007) rather than promote the book. 

I am uncertain whether this program should be promoted in this way. Is the paper therefore an evaluation of the PDEP. If so, what are the evaluation techniques that are in place? 

Rework the “Purpose of the study” so that there are clear research questions. 

There are gaps in this study – for example, why was there a focus on parents from marginalised communities? Surely there are so many other factors to consider if researchers are choosing vulnerable populations from “urban slums”. 

Given this marginalised community, provide greater detail on how sampling took place. 

A full copy of the survey should be included as well as the Flesch Reading Ease Test. 

Permission was obtained from the University of Manitoba for a study conducted in Bangladesh...this may require some defence. 

The Cronbach’s alpha scores for “Support of Physical Punishment” and “Parenting Self Efficacy” are low but these are not explained. 

I’m struggling to understand how a scale constructed in Canada was used to evaluate parent self-efficacy in Bangladesh. This is naturally going to result in high p values (significance scores). 

The Discussion is sparse and requires considerable amplification – the main results of the current study should be compared with other literature in the field.  

Reviewer 2 Report

Comments:

This paper examined the impact of a new intervention on parenting attitudes in a specific population. The study used the Positive Discipline in Everyday Parenting model developed in Sweden for a marginalized population in Bangladesh. The study had a decent research design and contributed to the existing literature on parenting in South Asian countries. Several comments are below.

1. Before discussing PDEP, you should have a literature review on the physical punishment of children. What are the negative effects? Why do we want to eliminate/reduce it? Then, you introduce the PDEP program.

2. Line 32: Check the reference formats. Is “Global Partnership to End Violence Against Children” the group name?

3. Line 44: Where is the Supreme Court, and what is the case?

4. Check the reference formats for internet sources.

5. Line 91-92: Need references.

6. Line 110-111: What are the percentages before the program? What is the statistical result? Is it significantly different before and after the program?

7. Line 116: List the countries in the sentence.

8. Line 123: Find references for this statement.

9. Lines 141-144: What is the sampling method in the current study?

10. Line 150: Is there government data for such a statement? If you don't have data, you will need to review studies on such community and their family situations.

11. Typos: Line 20 - "didn’t" should be "did not." Line 237- The font of the in-text citation does not match.

 Very interesting study. I would like to see a follow-up study.

Reviewer 3 Report

This paper is very important and constitutes an important field in scientific research, especially in Bangladesh, but there are two things that must be accomplished before acknowledging the acceptance of this research.

The first one is to use a social theory that analyzes behavior in this important topic, and the second is to conduct theory and previous studies in discussing the findings of this research.

I see that the discussion part is weak and needs more work, then the study can be accepted.

Round 2

Reviewer 1 Report

Thank you for developing the Discussion section which now offers a more animated dialogue for readers (both statisticians and non-statistician).

Reviewer 2 Report

The manuscript is improved significantly with the corrections.

Reviewer 3 Report

accept in present form